# Assessment of a 16-Channel Ambulatory Dry Electrode EEG for Remote Monitoring

**DOI:** 10.3390/s23073654

**Published:** 2023-03-31

**Authors:** Theeban Raj Shivaraja, Rabani Remli, Noorfazila Kamal, Wan Asyraf Wan Zaidi, Kalaivani Chellappan

**Affiliations:** 1Department of Electrical, Electronics and System Engineering, Faculty of Engineering and Built Environment, Universiti Kebangsaan Malaysia, Bangi 43600, Malaysia; 2Department of Medicine, Faculty of Medicine, Universiti Kebangsaan Malaysia, Cheras 56000, Malaysia; 3Hospital Canselor Tuanku Muhriz, Universiti Kebangsaan Malaysia, Cheras 56000, Malaysia

**Keywords:** EEG, wearable, remote monitoring, signal quality, epilepsy

## Abstract

Ambulatory EEGs began emerging in the healthcare industry over the years, setting a new norm for long-term monitoring services. The present devices in the market are neither meant for remote monitoring due to their technical complexity nor for meeting clinical setting needs in epilepsy patient monitoring. In this paper, we propose an ambulatory EEG device, OptiEEG, that has low setup complexity, for the remote EEG monitoring of epilepsy patients. OptiEEG’s signal quality was compared with a gold standard clinical device, Natus. The experiment between OptiEEG and Natus included three different tests: eye open/close (EOC); hyperventilation (HV); and photic stimulation (PS). Statistical and wavelet analysis of retrieved data were presented when evaluating the performance of OptiEEG. The SNR and PSNR of OptiEEG were slightly lower than Natus, but within an acceptable bound. The standard deviations of MSE for both devices were almost in a similar range for the three tests. The frequency band energy analysis is consistent between the two devices. A rhythmic slowdown of theta and delta was observed in HV, whereas photic driving was observed during PS in both devices. The results validated the performance of OptiEEG as an acceptable EEG device for remote monitoring away from clinical environments.

## 1. Introduction

Epilepsy is a class of chronic, non-communicable neurological conditions marked by recurrent spontaneous seizures [1,2]. According to the World Health Organization (WHO), approximately 50 million people from all walks of life are affected by epilepsy, putting it among the most prevalent neurological conditions worldwide [3]. Additionally, the WHO has also reported that with adequate diagnosis and treatment, up to 70% of epilepsy sufferers have the potential to achieve a seizure-free life [3]. Meanwhile, the increasing cost of healthcare, the pandemic, and limitation in access to healthcare services has further enhanced the trend of shifting healthcare services from clinics to the home [4]. Early in 2021, telehealth usage was 38 times greater than it was before the COVID-19 pandemic [5]. The emergence of a vast array of wearable technology in the field of neuro- and biotechnology, which provides real-time and continuous monitoring of physiologic as well as neurological activities, has been a catalyst in this shift towards remote healthcare [6,7,8,9,10,11,12]. The development of these wearable devices and smartphone applications has been identified as a clear step in assisting epilepsy patients to monitor the progression of their condition [13]. Usually, EEG tests are used to diagnose brain-related diseases, and they may help to pinpoint specific symptoms, such as seizures, and to identify seizure focus [14].

Currently, EEG is a crucial diagnostic tool for several neurological conditions, including epilepsy [15,16]. EEG signals are mostly low-voltage electrical impulses that are recorded by electrodes during brain activity and analyzing them can uncover critical human health issues [17]. In the investigation of seizure disorders, an EEG test aids in determining the type of seizure syndrome that patients are experiencing [18]. There are various types of EEG tests, including routine EEG, sleep EEG or sleep-deprived EEG, ambulatory EEG, video telemetry, and invasive EEG telemetry [19]. The current protocol used in the early diagnosis of epilepsy is a routine EEG test, which typically consists of a hyperventilation test, photic simulation test, and sleep test [20]. These steps in a routine EEG test are attempts to trigger a seizure while capturing the EEG signals.

Standard clinical EEGs have historically been somewhat expensive, had a limited range of motion for patients, and required a significant amount of time to prepare for. They are also usually bound to clinically controlled environments and are technologist dependent [21]. In general, home techniques are thought to be less expensive than in-patient EEG recordings. Moreover, at home, epilepsy patients may be subjected to their usual triggering circumstances, such as sleep deprivation [15]. In the past, commercially available wearable EEG devices have been studied for reliability in diagnostic usage [17,22]. However, the cost of the device can be a factor of obstruction in real-life applications of remote monitoring, as 80% of people with epilepsy are in low- and middle-income nations, according to WHO in 2022 [3].

There are several initiatives to solve remote monitoring needs using wearable devices. Despite the fast growth in digital health, EEG monitoring is still a challenge in remote monitoring due to its high cost and availability. Today, many researchers have chosen to develop new devices rather than using commercially available devices. In 2022, Gao et al. [23] investigated a new wearable EEG device’s signal quality for a brain computer interface (BCI). The device was constructed using an ADS1234 analog-to-digital converter (ADC) and had four dry electrode channels. However, the device is limited to frontal lobe application with only four electrodes available. Sintosky and Hinrichs [24], in 2020, devised an in-ear EEG for home monitoring. The device, which had a headphone design, was created to record biosignals within the ear canal using an ADS1299 ADC chip from Texas Instruments. The device is only applicable for single channel hearing-related biopotential capturing. In 2018, Lin et al. [25], created a single channel EEG with the purpose of detecting epileptic seizures in real-time. The system used an updated Open-RISC1200 processor core and a sigma-delta analog-to-digital converter (SD-ADC). The single-channel EEG was integrated into a headband for easy application. As per the clinical needs, a single channel will not serve the purpose to assist in diagnosis and treatment. In 2021, Valentin et al. [26] and Mai et al. [27], developed a custom-made EEG for on-the-go BCI acquisition and human emotion recognition, respectively. Both devices were based on Texas Instrument’s ADS1299 ADC which can provide up to 8 EEG channels with 24-bit resolution. An eight-channel EEG will limit the capacity of remote monitoring to a specific group rather than creating an opportunity for anyone clinically diagnosed with epilepsy.

The evolution of development in EEGs can be observed in numerous studies throughout the past; however, the trend highlights that a large portion of the studies has focused primarily on non-clinical applications, such as BCI and emotion monitoring [12,27,28,29,30,31,32]. Most of the devices have limited channels between one and eight. As such, advancement in mobile EEG for clinical-based remote monitoring for epilepsy treatment management is still in demand. The purpose of this study is to develop a home-based scalable 16-channel EEG device for the remote monitoring of epilepsy patients. This approach is expected to provide a patient-centric EEG device to aid in personalized epilepsy management with comprehensive monitoring systems.

## 2. Materials and Methods

### 2.1. EEG System Modelling

The proposed EEG device architecture was designed based on IoT key building blocks, as in Figure 1 [33]. There are four blocks in the device architecture design: (1) smart things, (2) network, (3) middleware, and (4) application. Smart things refer to the physical device and controller, which in this model is the EEG device. Network is the communication gateway that handles the transmission of data. Middleware is responsible for data storage and signal processing, while visualization is carried out in the application block.

The proposed EEG device’s hardware architecture was constructed with Cyton and Daisy DAQ boards, which are built upon ADS1299 ADC and PIC32MX250F128B microcontrollers. For a better understanding of our proposed EEG device architecture, we will briefly explain the ADC and microcontroller. ADS1299 is an 8-channel customizable ADC used for bioelectrical measurements, whereas PIC32MX250F128B is a 32-bit RISC CPU with low current consumption. The integrated circuit provides the necessary criteria for EEG measurements with 1 μVpp input noise, 110 dB CMRR, programmable gain, 24-bit data resolution, and up to 256 Hz of sampling frequency [34,35,36,37].

The development process of the EEG device is categorized into three different development groups: (1) hardware development, (2) software development, and (3) communication gateway development. Upon completion of all developments, device integration was performed to complete the device.

#### 2.1.1. Hardware Development

The EEG hardware was developed through an experimental method, which consisted of an exploratory study to select and develop a suitable device for remote monitoring. The hardware consists of data acquisition (DAQ) boards, headwear, and electrodes. The development process consists of: (1) DAQ board selection, (2) headwear development, and (3) electrode configuration.

The hardware development commenced with the exploration of DAQ boards in the market. A new board development was not considered as there were many types of DAQ boards commercially available in the market. In this study, we aimed to explore the limitations of current boards that are experienced by various developers in the field. As such, four different commercially available EEG boards were compared with a focus on six different aspects, as defined in Table 1. The decision of the board selection was prioritized on the number of channels, cost, and communication technology. The DAQ board was targeted to be ambulatory, as such wireless communication was preferred in the selection. Moreover, a scalable and minimum of 16 channels were essential in developing an EEG device with option to personalize. Finally, the cost of the DAQ board will play a crucial role in the overall cost of the device, as such the most cost-effective DAQ board in the market was also desirable.

The headwear is a helmet-shaped structure, created to hold the electrodes and the DAQ boards. The frame was fabricated with adaptations of 3D printing technology with an open-source design from OpenBCI. The 3D printing method applied was fused deposition modeling (FDM), where a Creality CR-6 SE 3D printer was used to print parts of the frame. Polylactic acid (PLA) filament, a bio-sourced and biodegradable substance, was used to print the pieces [42]. PLA was chosen since it is the most straightforward polymer to print with and has decent visual quality [43]. The parts were printed with a heated bed temperature of 70 °C and nozzle temperature of 200 °C. Figure 2a shows the 3D printed parts for a complete unit of headwear, printed using turquoise-colored PLA filament. The parts were then cleaned and assembled, as shown in Figure 2b.

The DAQ boards and electrodes were then positioned on the completed frame. Comb-type dry EEG electrodes composed of acrylonitrile butadiene styrene (ABS) plastic with a silver chloride (AgCl) coating (Figure 3a) were used. Unlike wet electrodes, dry electrodes do not require additional gels or conducting agents, which reduces the device setup time. The comb electrodes have 5 mm blunt prongs to accommodate longer hair while providing high signal quality and comfort to the user. The combs are attached to a wired holder (Figure 3b) which typically holds and connects the comb to the board, supported by a spring to provide sufficient pressure required to make feasible contact with the scalp. The holders are assembled by screwing them into the corresponding nodes available on the frame, the circular holes. Each node on the frame is an electrode location assigned based on the internationally recognized 10–20 system for electrode placement [44]. The EEG device runs on a rechargeable lithium polymer (LiPo) battery which can power the device for up to 20 h, suitable for long-term remote monitoring applications.

#### 2.1.2. Communication Gateway Development

The role of the communication gateway in device development is to manage data transmission and data storage. An agile method was applied in the development of the communication gateway to ensure the developed gateway met the requirements of the device in remote monitoring and to avoid further changes. The gateway device handles data transfer between the EEG hardware (host device) and the slave devices eliminates the need for bulky hardware, such as a laptop to monitor the EEG while recording. This further enhances the remote monitoring quality of the device. A Raspberry Pi 3 Model B+ (Rpi) was utilized to host a server. The server program was created by modifying a Raspberry Pi Buster operating system. It is a lightweight program and runs as a background daemon upon powering ‘On’ the Rpi. The server can connect to the DAQ boards for wireless data receiving and transmission. The EEG hardware transmits data through Bluetooth to a USB dongle, which is connected to the USB port of the Rpi. The server receives data from the EEG hardware and streams it to be viewed from client devices.

#### 2.1.3. Software Integration

Digital lifestyles today focus on mobile applications, and continuous growth in the field has paved way for an endless possibility of applications. A mobile application can be an ideal replacement for visualization and control hardware, such as a laptop to monitor, and can control the EEG device. As such, a mobile application was obtained and integrated into the EEG device. The application is an Android-based app that enables users to view real-time EEG data and record it when necessary.

#### 2.1.4. Device Integration

Device integration refers to the integration of the EEG hardware, mobile application, and communication gateway. The integration is necessary to allow interoperability of EEG data sourcing, recording, and visualization. The overall architecture of the integrated system is shown in Figure 4. Real-time EEG signals are transmitted from the hardware to the Rpi (local server). The local server receives the data and streams it while also saving the data locally in the memory card of RPi. The streamed data is viewed from the mobile application as a client. The future idea is to store the captured data in the cloud for further analysis and analytic reports.

### 2.2. Device Verification

Device verification is a standard practice in the process of device development to ensure that the outputs match the user requirements. A series of lab tests were conducted to ensure the ability of the device in recording quality EEG signals. In addition to the signal quality assessment, a reproducibility test was conducted, as advised by a neurologist. The proposed reproducibility test was conducted at the Neurology Lab of Hospital Canselor Tuanku Muhriz (HCTM), UKM. This test was conducted to assess the ability of the device to respond to the clinical EEG routine test. The reproducibility test included preparation and a series of tests as shown in Figure 5. The test was conducted under the supervision of a neurologist and a technologist.

I.EEG Setup and Subject Preparation

The test was conducted on a 30-year-old healthy male with no history of an epileptic seizures. Medium-sized headwear was chosen based on the subject’s head circumference of 55 cm. The EEG device was placed on the subject’s head while the subject was in a seated position, and the electrode units were gradually tightened until the electrode tips touched the scalp. An impedance check was carried out to ensure that every electrode relates to an acceptable impedance threshold, which in this study was set below 2000 ohms.

II.Eye Blinking Test

The eye blinking test was conducted to test sensitivity of the EEG device in detecting eye blinking artifacts. The subject was instructed to blink once and then blink rapidly while in a seated position.

III.Eye Open and Close Test

The eye open and close test was conducted to verify the ability of the device in obtaining EEG signals during eye opened and closed states. For this test, the subject was instructed to repeatedly open and close his eyes at an interval of 10 s for 10 times, to study the effect of the stimuli.

IV. Photic Simulation Test

The photic stimulation test was conducted to verify the ability of the device in obtaining signals during photic simulation. A light source was shifted toward the subject and carefully placed above the eye line about 50 cm from the eyes, while the subject was in a seated position. The subject was instructed to keep his eyes closed before and after flashing followed by flashing in the eye open condition at a frequency of 1 Hz for 10 s. The procedure was repeated with different frequencies of 3 Hz, 6 Hz, 8 Hz, 10 Hz, 12 Hz, 14 Hz, 16 Hz, 18 Hz, 20 Hz, 25 Hz, and 30 Hz.

V. Sleep Test

The sleep test was conducted to verify the comfortability of the device while in a sleeping position. The subject was instructed to lie in a supine position and to sleep during the recording.

### 2.3. Clinical Device Validation

Device validation is vital in assessing any device, especially a medical device intended for a selected clinical procedure. In this study, signal quality of the EEG device was compared to a conventional, clinical EEG device at HCTM. Thirteen control subjects were recruited from various backgrounds for the experiment. Participants were made aware of the details and procedures of the study, and then participants’ written consent was obtained. This study was approved by the Research Ethics Committee of UKM (JEP-2022-430). The clinical EEG device used was the NicoletOne EEG System, Natus.

#### 2.3.1. Signal Quality Experiment Design

A routine EEG test at HCTM consists of four tests, namely the eye open and close (EOC) test, hyperventilation (HV) test, photic stimulation (PS) test, and sleep test. Three of these tests were conducted on the subjects with both devices, with the sleep test avoided due to the time factor. EEG signals from Natus were collected using gold cup gel electrodes, while OptiEEG used dry electrodes. A total of 21 channels were utilized for Natus, whereas for OptiEEG 16 channels were used, both based on the 10–20 system. The subjects had different head circumferences and, as such, suitable sizes of OptiEEG were assigned accordingly. Prior to the experiment, two medium and one large size units of OptiEEG were fabricated.

OptiEEG employed a sampling rate of 125 Hz, while Natus used a sampling rate of 500 Hz. In this experiment, a similar recording setup (environment and body position) was applied for both devices, where the subject laid on a bed with the upper body slanted about 45° upwards. The room was also dimly lit to avoid interference from external light sources.

#### 2.3.2. EEG Pre-Processing

The EEG data collected in the clinical experiment was pre-processed using MATLAB for further analysis. The steps taken in pre-processing are as follows:

Step 1: Raw EEG signals collected from Natus and OptiEEG are segmented according to each of the tests, namely the EOC test, HV test, and PS test.

Step 2: The segments are then filtered with a Butterworth bandpass filter of order 5 (0.5 Hz and 40 Hz).

Step 3: The filtered signals are then normalized for standardization between the two devices.

#### 2.3.3. Statistical Analysis

The EEG data collected from both devices were compared by computing signal-to-noise ratio (SNR), peak signal-to-noise ratio (PSNR), and mean square error (MSE) using MATLAB. In this study, noise, N, was classified as power frequency noise, which consists of baseline noise (below 0.5 Hz) and high-frequency noise (above 40 Hz). The clean EEG signal, *b*, is the signal obtained after filtering and normalizing the raw EEG signal, *a* [45]. The SNR which represents the power ratio of the clean EEG signal and the noise is computed based on this assumption. MSE is calculated by dividing the squared norm of the difference between the data and the approximation by the number of elements, *n*. PSNR is the peak signal-to-noise ratio, measured in decibels [28,46,47]. The SNR, MSE, and PSNR were computed based on the following formulas:(1)$$\mathrm{SNR}=10\times \log_{10}\left(\frac{P_{a}}{P_{N}}\right).\;$$
(2)$$\mathrm{MSE}=\frac{1}{n}\sum_{i=1}^{n}{\left(a-b\right)}^{2}$$
(3)$$\mathrm{PSNR}=20\log_{10}\left(\frac{MAX_{f}}{\sqrt{\mathrm{MSE}}}\right)$$
where *a* is the original signal, *b* is the cleaned signal, *n* is number of samples, *P_a_* is power of the original signal, *P_n_* is the power of noise, and *MAX_f_* is the maximum signal value that exists in the cleaned signal.

#### 2.3.4. Wavelet Analysis

Decomposing non-stationary EEG data into time–frequency sub-bands and then analyzing the signals is a common method of processing non-stationary EEG signals [48,49]. Wavelet transform simultaneously provides time and frequency viewing of a signal, allowing exact capture and localization of transient features, such as epileptic spikes in the data [50]. We have decomposed the collected data using discrete wavelet transform (DWT) [51] and adopted Daubechies as the mother wavelet [52,53]. Five frequency sub-bands were extracted from each signal based on clinical interest, that is delta (0–4 Hz), theta (4–8 Hz), alpha (8–12 Hz), beta (13–30 Hz), and gamma (30–50 Hz) [48,54,55,56]. Table 2 shows the level of decompositions for Natus and OptiEEG EEG signals, and the corresponding frequency band. The energy of selected frequency bands was then computed.

## 3. Results

The proposed EEG device was fabricated for the remote monitoring of epilepsy patients. As such, the device was targeted to be ambulatory and optimized with low set-up complexity to allow the patients to use the device with minimum training. In this section, details and functions of the developed device are presented.

### 3.1. EEG Device

The market study conducted on DAQ boards resulted in opting for OpenBCI’s Cyton and Daisy EEG boards, as they correspond to the preferred requirements while being cost-effective. The two DAQ boards are connected end-to-end, resulting in a 16-channel amplifier. Three complete units of the EEG device were developed as shown in Figure 6.

The EEG device can record up to 16 channels of EEG signals. Real-time EEG can be visually monitored using the developed mobile application or the OpenBCI Graphical User Interface (GUI). Figure 7 below shows a screenshot of 16-channel real-time EEG obtained from the device using OpenBCI GUI. The recorded EEG can be saved directly on a memory card that is fitted into the DAQ board, in the Raspberry pi, or in the computer while recording.

The app is features the ability to display real-time raw or filtered EEG data with a variable filter option. Other features include the option to start or stop EEG recording, view the EEG band in real-time, and shut down the EEG device. The ‘Filtered EXG’ screen displays real-time filtered EEG for an 8-channel configuration as shown in Figure 8a. There is a built-in filtering option to choose the frequency of the notch filter that will be applied to the acquired raw EEG signal in order to visualize the raw data in a graphical format, as in Figure 8a. Altering the graph’s vertical (amplitude) and horizontal (time) scales is another option. The ‘Raw Data’ screen is depicted in Figure 8b, where the device’s real-time raw EEG data can be viewed in numeric form and the recording can be started or stopped as well. The recording’s preferred name can also be specified by users. The ‘Band Power’ screen displays the real-time band power of the channels, as shown in Figure 8c.

### 3.2. Device Verification Observations

The device verification session was carried out in the Neurological Unit, in the presence of a neurologist with a specialty in electrophysiology and epileptology. The test was conducted in the standard clinical setting for any EEG recording for epilepsy patients. The following are the test observations:I.Eye Blinking Test

The results of this test were observed in the frontal lobe (Fp1 and Fp2). The neurologist verified the presence of eye blinking artifacts in the frontal electrode for a single blink (Figure 9a) and rapid blink (Figure 9b), indicating the device’s sensitivity to test stimuli.

II.Eye Open and Close Test

The response of this test was observed in the occipital lobe (O1 and O2). The neurologist verified that the alpha wave is dominant during the ‘eye close’ state, indicating that the hardware manages to detect changes in EEG signals between opened eyes (Figure 10a) and closed eyes (Figure 10b).

III.Photic Stimulation Test

The observation of this test was found to be in the occipital region as well. During PS, a widespread rhythmic activity that follows the frequency of the flickering light was observed. Figure 11 shows the EEG signals of the occipital region before photic (Figure 11a) and during photic stimulation (Figure 11b).

IV. Sleep Test

The patient was unable to fall asleep during the assessment; hence, the sleep test was not conducted.

### 3.3. Device Validation Results

This section includes responses of the OptiEEG device signal analysis in comparison to Natus, the EEG device used in HCTM’s Neurological Unit clinical setting.

#### 3.3.1. Statistical Analysis

Data from the two different devices were compared visually for the response from the clinical epilepsy EEG patient routine assessment. These observations were conducted based on the neurologist’s request to evaluate the routine assessment response in OptiEEG. The OptiEEG and Natus EEG devices in the neurological clinic were used for the recording. The figure below presents recorded signals of OptiEEG (Figure 12a) and Natus (Figure 12b), which represent 16 channels each. As observed, the recorded signals had a good degree of accuracy and sensitivity.

The routine EEG test time includes two phases, namely the device preparation time (OptiEEG: 12 min, Natus: 28 min) and the recording time, varied between 12 to 14 min for OptiEEG and Natus, which includes the EOC test, HV test, and PS test. In reference to current practice in the clinical setting using a gold cup and gel, the proposed dry electrode has reduced 50% of the setting time. Gold cup setting needs trained technologists, whereas the proposed dry electrode can be managed by patients themselves or family members with minimum training. In terms of complexity, the patient’s involvement in electrode setting is just to ensure the electrode is tightened to touch the scalp since the EEG headwear has been designed to comply with the 10–20 system and sized to the patient’s head size. The second part focuses on raw EEG signals acquired from both devices, which were segmented according to the length of each test, and each segment was pre-processed to compute SNR, PSNR, and MSE. The segmented signals were filtered using the Butterworth bandpass filter of order 5, with 0.5 Hz and 40 Hz cutoff frequencies.

The SNR, PSNR, and MSE assessment that was conducted for both OptiEEG and Natus’s recorded signals were found to have a very low standard deviation, as stated in Table 3. A lower standard deviation indicates less variation in the recorded 13 subjects’ signals in both devices. The SNR and PSNR analysis also identified that there is a variation in Natus channel values compared to OptiEEG, which has consistent SNR and PSNR values across the channels. This difference is due to the reference position in these two devices. OptiEEG was referenced at A1 and A2, positioned at the ear, whereas Natus was referenced at Cz, according to the 10–20 electrode placement system [44,57,58,59]. The SNR and PSNR of OptiEEG were slightly lower than for Natus, but within an acceptable bound. The standard deviations of MSE for both devices are almost in the similar range for all three different tests.

#### 3.3.2. Wavelet Analysis

Wavelet decompositions were performed on signals recorded from both devices for three different routine EEG clinical tests. The first two activities were observing physiological responses, whereas the third activity was observing an external triggering factor. The past literature has indicated that the selected three routine test responses are mostly observed in the occipital region that is represented by channel O1 and O2 in the 10–20 electrode placement system [60,61,62].

A rhythmic EOC test with intervals of 10 s was recorded for a duration of 4 min in both devices. Lal Hussain et al. [63] have stated that values of eyes closed and eyes open of subjects over 19 channels were obvious at occipital electrodes, and there is a clear depiction between eye open and closed even though the differentiating threshold is very small. In this study, the researchers have found that the energies of the different frequency bands are consistent between the two devices, as in Figure 13.

HV is a standard activation method practiced in routine EEG recording among epilepsy patients [64]. HV is a condition where a person breathes faster and deeper, which can lead to changes in brain activity, including a bifrontal preponderance pattern. The activation technique generally results in a physiological slowing of brain rhythms. The typical HV response in EEG shows moderate to high voltage, often rhythmic with delta and theta slowing in conjunction with bifrontal preponderance [61]. Bifrontal preponderance is a term used in EEG to describe an abnormality in the distribution of electrical activity in the brain. Specifically, it refers to an excessive amount of activity in the frontal lobes of the brain relative to the posterior regions. In this study, the HV test is used as a validation method to verify the capacity of OptiEEG in capturing HV response in comparison to the Natus EEG device, as in Figure 14.

The next common activation technique used in a routine EEG is PS. PS induces photic drive, a rhythmic frequency in the occipital derivations that consists of harmonically related activity to the flickering light [61]. Photic driving involves synchronization of the occipital alpha rhythm with the stimulus frequency [60]. The study also used this test to verify the capacity of OptiEEG response towards capturing the photic rhythmic response from Natus, as in Figure 15.

## 4. Discussion

In the past few decades, EEG recordings and assessments in epilepsy treatment monitoring have only focused on clinical settings, even though Penry and Dreifuss discussed the significance of long-term EEG monitoring among seizure patients way back in 1969 [65]. The advancement of research and the recent pandemic challenge has introduced a new trend in disease management and monitoring. This study is proposing a potential home-based EEG monitoring solution using dry electrodes with a personalized 10–20 electrode placement system that provides self-setting facilities. In more detail, we aim to personalize the headwear using additive manufacturing technology and a scalable channel setting based on the clinical assessment. The need for ambulatory EEGs was identified as early as 1962 using radio telemetry [66], and in 1975 a 4-channel 24-h cassette recorder was used for long-term recording, yet EEG tests were often performed during working hours, and it was impractical for patients who had attacks outside of regular business hours [67]. In 1979, commercial use of cassette tape recorder-based ambulatory EEG monitoring began [68]. It held considerable potential for aiding in the differential diagnosis of episodic loss or alteration of consciousness by offering portable, outpatient and inpatient enhanced EEG temporal sampling and other physiological parameters. Patients were also allowed to engage in their daily activities without the requirement to be hospitalized. Other benefits were the ability to recreate the relevant occurrences in a natural home or work environment with the accompanying typical stress [68].

Remote monitoring medical devices are also known as wearable sensing instruments that are growing in reputation both in general physiological monitoring and for health applications, such as seizure monitoring. However, there are limited data about these devices’ reliability [69]. In addition to these challenges, there are several separate issues to be considered, such as data transmission and availability. The OptiEEG device developed in this study is complemented by IoT technology. The IoT module comprises a Raspberry Pi for storage and a mobile application as a visualizer.

The 16-channel dry electrode OptiEEG is expected to cater to the usability challenges faced in most of the present EEG devices, including the ones in the clinical setting. The need for a dedicated neurotechnologist to set up the devices and the time taken for the setting has been eliminated in OptiEEG. OptiEEG is designed using a 10–20 electrode placement system that is customizable for the client through additive manufacturing technology. OptiEEG also provides a more comfortable option by using dry electrodes instead of wet electrodes that need more preparation time and eliminates the drying gel limitations in long-term recordings. Even though the electrode has a significant advantage in terms of patient preparation, there is a substantial gap identified in the OptiEEG headwear design that is limiting patients’ comfort during recording in a supine position. This limitation is suspected to contribute to the high gamma and beta values in some of the recordings. The next phase of the study is expected to explore a better headwear design that allows more comfortable supine position recording. In 2022, Andrea Biondi et al. have carried out an extensive review on the use of noninvasive mobile EEG for seizure monitoring and management. The study comprehensively reviewed the different components of the EEG device that includes electrodes and electrode placement, battery, sampling rate, resolution, data transmission, seizure detection algorithm, and the support team availability. This study concluded that when seizure detection and management are performed remotely, there is a high potential for change in terms of time, technical assistance, cost, usability, and reliability [70].

In this study, we have carried out clinical studies to verify and validate the reliability and usability of the device against the clinical device at HCTM. A total of 13 subjects have undergone recording from both devices using the routine EEG tests, namely EOC, HV, and PS tests. As discussed in the results, the outcome of both recordings was assessed for their reliability in terms of data quality using SNR, PSNR, and MSE. The average standard deviation of the SNR, PSNR, and MSE for OptiEEG is below 0.5 compared to Natus (<1.5). Further assessment was carried out on the energy of the frequency bands between the two devices. Figure 13, Figure 14 and Figure 15 indicate a similar pattern of the energy bands in the three different routine EEG tests conducted.

Currently, there are various devices that can be applied for EEG monitoring, as detailed in Table 4. Among the different monitoring devices, g.tec devices are one of the most used for EEG [71]. Compared with g.tec, the developed EEG monitoring system has better flexibility to be used for remote monitoring among epileptic patients due to the simplicity in setting up and the customization of the electrode channels. The user can use the device in a home setting without the assistance of an expert. The device is designed with local storage capacity in the absence of wireless communication support, which enables monitoring to be carried out in remote rural locations. Thus, compared with most EEG devices in the market, this system can be customized for home-based monitoring both with and without wireless infrastructure, that accommodates the needs of rural locations without sufficient communication infrastructures.

In future work, we aim to evaluate OptiEEG’s performance among epilepsy patients with general and focal epilepsies. A revised test protocol to include 20 min of resting between the three routine test procedures is recommended. This recommendation is based on the limitations faced in the current statistical data analysis. Even though the data recorded using the present protocol produced reliable outcomes in signal quality, it limits the options in advanced data analytics to explore the reliability and usability of the data for clinical decision-making and monitoring. The research team is also considering a machine learning or deep learning approach to assist in seizure presence detection [75] from all the 16 channels and to categorize the severity of the identified seizure. As part of phase two of the device testing and design review process, this study also includes the device certification initiative and intellectual property (IP) filling to ensure OptiEEG would be clinically applicable.

## 5. Limitations

The study presented in this manuscript is focused on measuring the dynamics of the EEG recordings of the two devices, namely OptiEEG and Natus. Three different tests were conducted to indicate the signal state change. The study produced interesting findings, but it has several gaps, which include a small sample size and lack of analysis, such as by gender and age. Furthermore, at this stage of the study, no epileptic subjects are recorded for analysis. Future research can be designed to quantify the dynamic reliability for a larger group based on healthy and epileptic subjects, considered with age and gender as well, to ensure the reliability of OptiEEG over a wider range of users.

## 6. Conclusions

In this study, a practical wearable 16-channel EEG remote monitoring device, OptiEEG, has been proposed, and a comprehensive assessment of the signal quality was carried out on OptiEEG in comparison to Natus, a clinically approved EEG device. In addition, we further verify OptiEEG’s feasibility in clinical simulation applications to track the attention states. The simulated signal test results demonstrated that OptiEEG circuit is compatible, and the collected EEG data reflects a similar activity to Natus. Another test comparing OptiEEG’s signal quality to that of the industry’s gold standard for EEG acquisition, Natus, revealed that OptiEEG can replicate Natus’s results. Based on EEG data of eye open/close, hyperventilation, and photic stimulation tests, the consistent signal quality of OptiEEG was proved by the SNR, PSNR, and MSE. The above results provide evidence that the OptiEEG as a monitoring device can reliably record the different simulation tests that have been in practice to trigger events among epilepsy patients during clinical visits. The obtained results have provided the baseline indication of the OptiEEG’s reliability to be used as a home-based monitoring device for epilepsy patients.

## Figures and Tables

**Figure 1 sensors-23-03654-f001:**
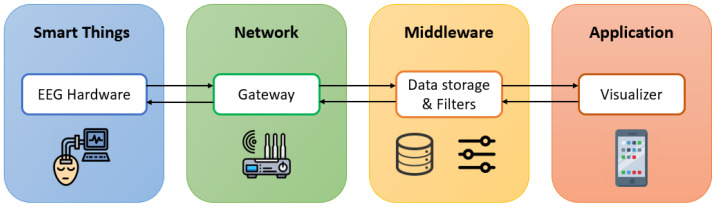
Proposed EEG system architecture.

**Figure 2 sensors-23-03654-f002:**
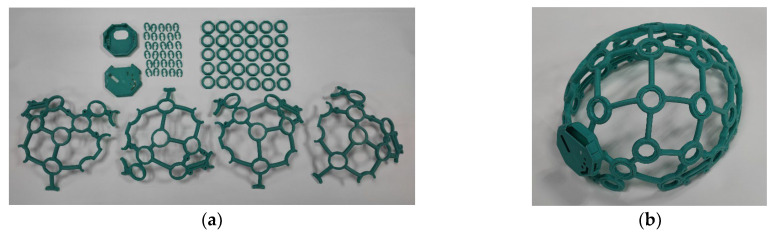
3D printed headwear. (**a**) Printed parts; (**b**) assembled headwear.

**Figure 3 sensors-23-03654-f003:**
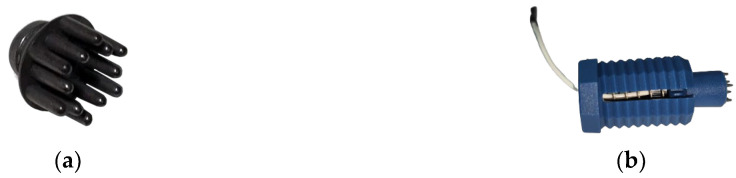
Dry electrode. (**a**) Dry comb electrode. (**b**) Wired holder.

**Figure 4 sensors-23-03654-f004:**
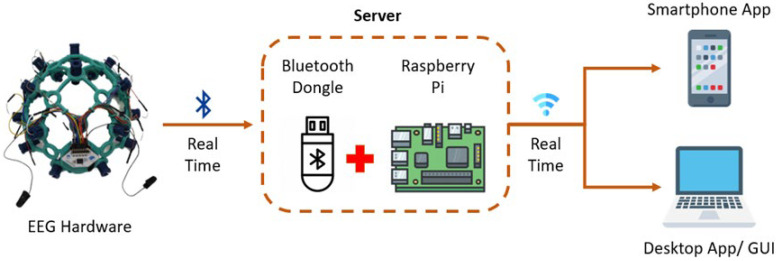
EEG device architecture.

**Figure 5 sensors-23-03654-f005:**
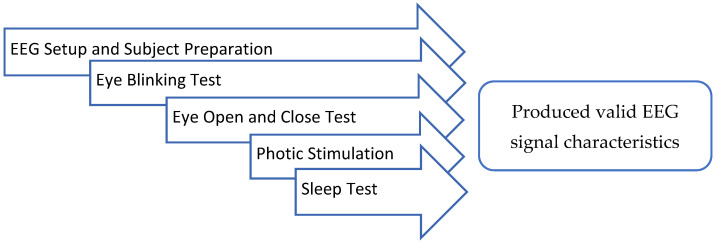
Reproducibility tests.

**Figure 6 sensors-23-03654-f006:**
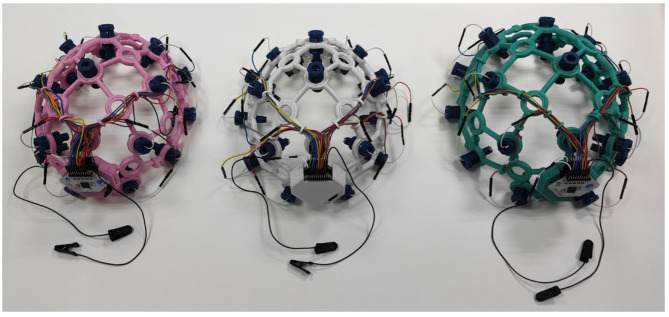
EEG devices.

**Figure 7 sensors-23-03654-f007:**
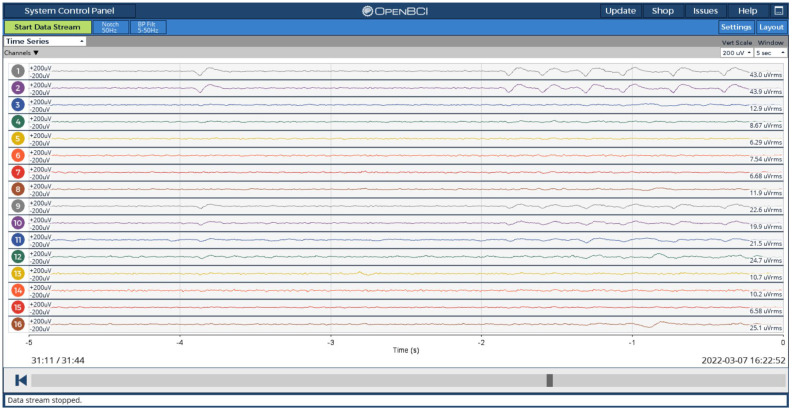
16-channel real-time EEG.

**Figure 8 sensors-23-03654-f008:**
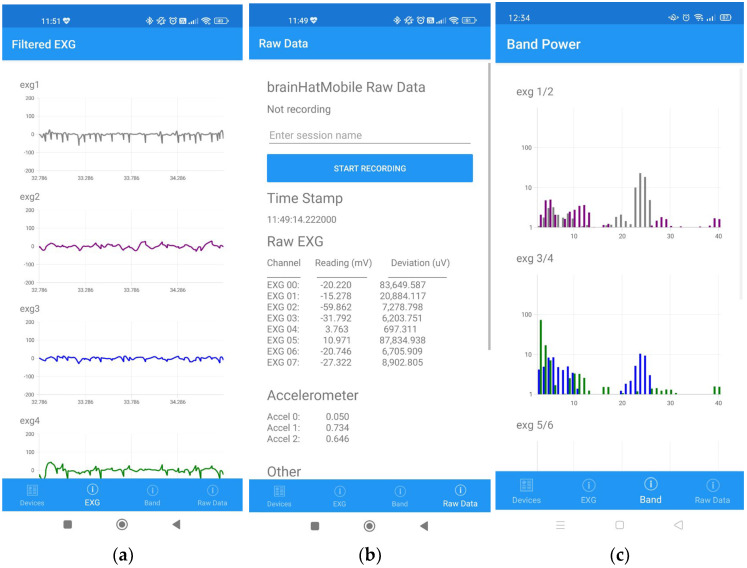
Mobile application. (**a**) Filtered EXG screen. (**b**) Raw Data screen. (**c**) Band Power screen.

**Figure 9 sensors-23-03654-f009:**
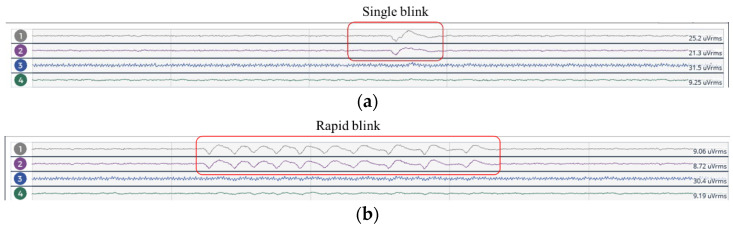
EEG response to the eye blinking test. (**a**) Single blink; (**b**) rapid blink.

**Figure 10 sensors-23-03654-f010:**
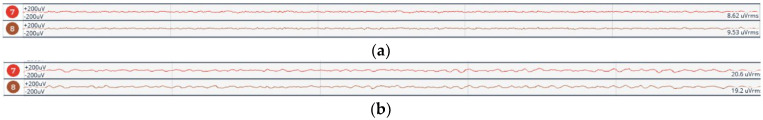
Response of occipital region during eye open and closed state. (**a**) Opened eyes; (**b**) closed eyes.

**Figure 11 sensors-23-03654-f011:**
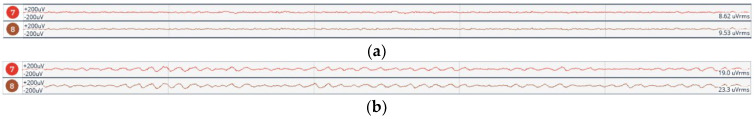
Rhythmic activity during photic simulation. (**a**) Non-photic; (**b**) photic.

**Figure 12 sensors-23-03654-f012:**
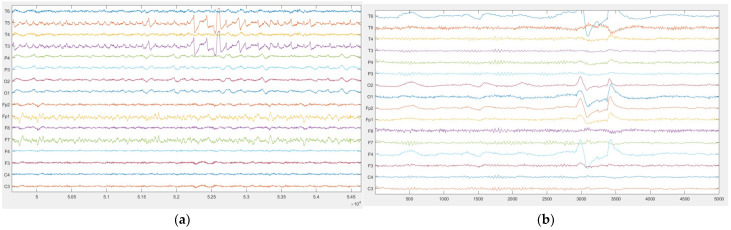
Recorded signals. (**a**) OptiEEG; (**b**) Natus.

**Figure 13 sensors-23-03654-f013:**
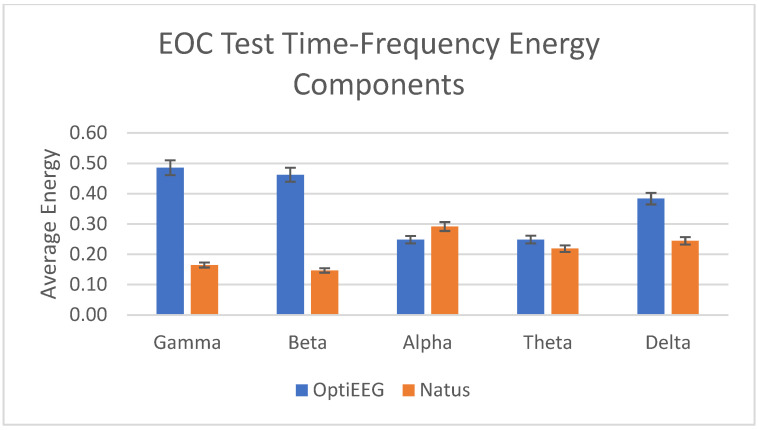
Power of time–frequency components for EOC test.

**Figure 14 sensors-23-03654-f014:**
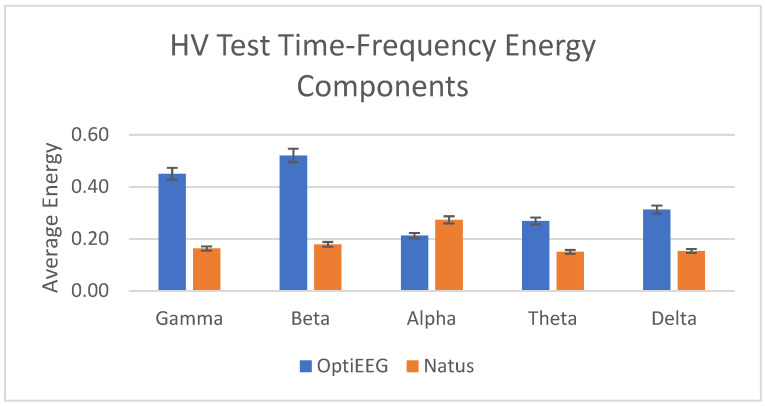
Power of time–frequency components for HV test.

**Figure 15 sensors-23-03654-f015:**
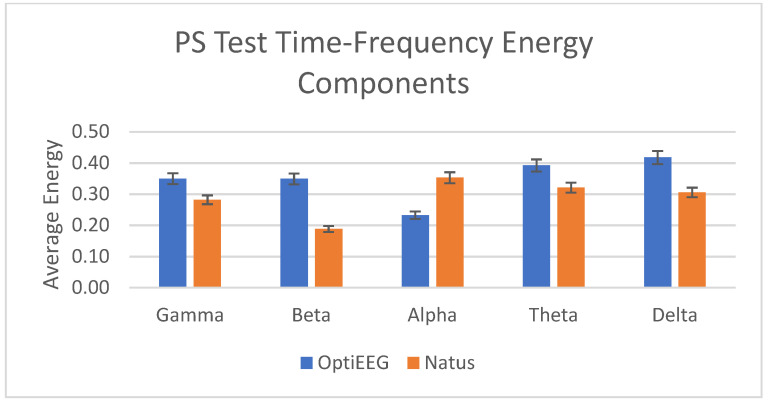
Power of time–frequency components for PS test.

**Table 1 sensors-23-03654-t001:** EEG DAQ boards comparison.

Features/Device	HackEEG	Texas Instrument ADS1299EEGFE	WallySci E3K	OpenBCI Cyton and Daisy
EEG Channel	8 to 32	8	6	8 to 16
Sampling Rate (Hz)	4000	250 to 16,000	1 to 2000	125 or 256
Connectivity	USB	USB	Bluetooth	Bluetooth
Raw EEG Data Access	Yes	Yes	Yes	Yes
Wireless	No	No	Yes	Yes
Cost	High	Low	Low	Low
Reference	[38]	[39]	[40]	[41]

**Table 2 sensors-23-03654-t002:** DWT decomposition levels and corresponding frequency bands.

Frequency Band	Frequency (Hz)	OptiEEG Decomposition Level	Natus Decomposition Level
Gamma	31.3–62.5	Detail 2	Detail 4
Beta	15.6–31.3	Detail 3	Detail 5
Alpha	7.8–15.6	Detail 4	Detail 6
Theta	3.9–7.8	Detail 5	Detail 7
Delta	0–3.9	Approximation 5	Approximation 7

**Table 3 sensors-23-03654-t003:** Signal analytics results.

		OptiEEG			Natus		
		SNR	PSNR	MSE	SNR	PSNR	MSE
Eye Open/close	Average	−2.74	−2.74	1.88	−0.19	−0.19	1.07
	Standard Deviation	0.24	0.24	0.12	0.71	0.71	0.17
Hyperventilation	Average	−2.48	−2.48	1.78	1.04	1.04	0.88
	Standard Deviation	0.46	0.46	0.19	1.26	1.26	0.21
Photic	Average	−2.71	−2.71	1.86	−0.05	-0.05	1.05
	Standard Deviation	0.29	0.29	0.14	0.96	0.96	0.20

**Table 4 sensors-23-03654-t004:** Comparison of ambulatory devices.

Key Feature	g.Nautilus PRO 16 g.SAHARA	SAGA (TMSI)	B-Alert X-Series	OptiEEG
Type	Ambulatory	Ambulatory	Ambulatory	Ambulatory
Input Channel	16	32	20	16
Setup Time	Fast	Slow	Slow	Fast
Operator Dependency	Not Dependent	Dependent	Dependent	Not Dependent
Electrode Positions	Pre-set 10–20 System	Pre-set 10–20 System	Requires measurement	Pre-set 10–20 System
Electrode Type	Hybrid (Dry and Gel)	Gel	Dry	Dry
Connectivity	Wireless (Bluetooth)	Wireless (Bluetooth)	Wireless (Bluetooth)	Wireless (Bluetooth)
Battery Life	10 h	6 h	8 h	20 h
Cost (US$)	12,600	32,377	14,950	2520
Reference	[72]	[73]	[74]	Proposed

## Data Availability

The data presented in this study are available on request from the corresponding author. The data are not publicly available due to ethical reasons.

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
