# Peer review of "Assessment of a 16-Channel Ambulatory Dry Electrode EEG for Remote Monitoring"

_sensors, 2023, doi:10.3390/s23073654_

Round 1

Reviewer 1 Report

The manuscript is a good initiative towards ambulatory EEG for seizure patients and very well written. I have some queries regarding this: 1. The size and structure of the electrode cap. How difficult is it for a patient to wear this cap? 2. The size of the data is too small and not taken from the seizure patients 3. The acquisition device and electrodes used are readily available. The only new thing I can notice is the 3D printed electrode cap 4. The comparison of the two models, proposed OptiEEG and Natus, is performed on two different signals as can be seen in Figure 17 5. What is the use of a notch filter in an application developed when the signal is filtered in 0-40Hz?

7) Authors have mentioned about reliability results however, other parameters such as real time setup time, speed of set-up, comfort in wearing the headband are not compared with already available headbands. 

8) A tabular comparison with 3-4 existing available headbands is required. 

Author Response

Dear Reviewer,

Thank you very much for your valuable input on our manuscript. We really appreciate every single feedback shared with us for the improvement of the article. This exercise has provided us an opportunity to enhance our research insides.

Following are the list of review outcome that has been addressed to the best of our understanding. Appreciate further assistance if there is further improvement required.

Regards,

Kalaivani Chellappan

Reviewer 2 Report

Mobile EEG recording in the home environment gained new interest due to technological progress and new impulses for telemedicine during the pandemics.

Thus, the presented system adds new options for remote monitoring of brain activity especially in patients with epilepsy.

Overall the manuscript is well organized and clearly written. Probably it can be shortened by combining or removing some of the figures. For example figure 8 can be omitted and replaced by a table giving the frequency ranges for the single scales. Also the amount of screenshots from the mobile app can be reduced or combined to a fewer number of figures.

Major points to reconsider are:  

1.    The manuscript is very detailed with respect to many components and parts of the device. But there is no information given about the dry electrodes. Geometry, surface material, etc.. This is of highest importance for the signal quality.

2.    It should be discussed, whether it is aimed to certify the device as a medical device. This is of high importance to really achieve clinical use.

3.    Figure 18 to 20 show higher power for OptiEEG for the beta and gamma range. Is it just noise picked up by the system or really oscillatory brain activity in these frequency ranges ? In the first case it should be discussed how the system can be made less susceptible to high frequency noise.

Minor:

1.    In the introduction it is stated: “The current protocol used in the early diagnosis of epilepsy is a routine EEG test, which typically consists of eye open and close test, hyperventilation test, photic simulation test, and sleep test [19,20]. These steps in a routine EEG test are attempts to trigger a seizure while capturing the EEG signals.

Recording with eyes open and closed is not performed to provoke a seizure but to show the Berger effect in the patient (occurrence of occipital alpha, when closing the eyes).

2.    Figure 7.: Though the subject agreed on her photos being presented in the journal, one could suggest to blurr the face or cover it with a circle to prevent the recognition of the person.

3.    There is a recent review on mobile EEG recording, which should be discussed in the manuscript:  Biondi, A. et al. (2021) ‘Remote and long-term self-monitoring of electroencephalographic and noninvasive measurable variables at home in patients with epilepsy (EEG@HOME): Protocol for an observational study’, JMIR Research Protocols, 10(3). doi: 10.2196/25309. The same group performs a structured home monitoring study in the UK: Biondi, A. et al. (2022) ‘Noninvasive mobile EEG as a tool for seizure monitoring and management: A systematic review’, Epilepsia, (December 2021), pp. 1–23. doi: 10.1111/epi.17220.

4.    Figure 18. Here it would be better to compare the both conditions (eyes closed vs. eyes open), which should show increased alpha power for the eyes closed condition.

Author Response

(The authors gave the same response as above.)

Reviewer 3 Report

The manuscript worth of publication, but not in its current presentation format. The manuscript is long and can be reduced by 30 to 40%.  The reader will get confused by the current quality of presentation. Please revise the manuscript so it can fit in 15 pages. The authors can ignore engineering aspect of the problem and focus on the research paper and what's novel in their solution.  In addition, the manuscript cites more than 60 references, which are not all relevant to this study, and ignore some important recent work in this field. The first equation is unnecessary. The limitation section needs major revisions. This section should highlight  the limitation of the current study in the context of related work.  I suggest accept with major revision.

Author Response

(The authors gave the same response as above.)

Round 2

Reviewer 3 Report

The authors did a good job in revising the manuscript. However, the paper is too long and very confusing. I suggest that the authors revise their manuscript and further shorten the paper. The paper should be shorten at least to 20 pages. 

The quality of some figures is very low that makes it hard for the reader. It would be acceptable to have a lengthy manuscript if it was a review paper, but this is a research paper and the engineering aspect is very dominating. Some figures are unnecessary and just add to the length of the paper. If the authors feel they are, they can put them as appendix and the reader can refer to them. 

The authors need to discuss how the platform can integrate machine learning and deep learning for real-time classification/detection. The authors can refer to the work of Raj Shekhar et al. https://www.mdpi.com/1424-8220/22/5/1852 or Ahmed et al https://www.frontiersin.org/articles/10.3389/fncom.2021.650050/full

The limitation of the proposed platform needs to be improved. 

The authors need to blur completely the face of the patient in the picture. OR Just remove the picture. It does not really add any value to the manuscript.

The authors need to discuss the biocompatibility of the 3D printout and the safety of the patient while using the platform.

Can the authors group figures 13, 14, 15? 

Author Response

Dear Reviewer,

Thank you very much for your valuable input on our manuscript. We really appreciate every single feedback shared with us for the improvement of the article. This exercise has provided us an opportunity to enhance our research insides.

Attached is the list of review outcome that has been addressed to the best of our understanding.

Appreciate further assistance if there is further improvement required.

Thank you.

Sincerely,

Kalaivani Chellappan
